# Efficiency of Orexin-A for Inflammatory Flare and Mucosal Healing in Experimental Colitis: Comparison with the Anti-TNF Alpha Infliximab

**DOI:** 10.3390/ijms24119554

**Published:** 2023-05-31

**Authors:** Anne Blais, Annaïg Lan, François Blachier, Robert Benamouzig, Pauline Jouet, Alain Couvineau

**Affiliations:** 1UMR-PNCA, Université Paris-Saclay, AgroParisTech, INRAE, 91120 Palaiseau, France; annaig.lan@agroapristech.fr (A.L.); francois.blachier@agroparistech.fr (F.B.); robert.benamouzig@aphp.fr (R.B.); 2Hôpital Avicenne, Assistance Publique-Hôpitaux de Paris, 93000 Bobigny, France; pauline.jouet@aphp.fr; 3INSERM UMR 1149/Centre de Recherche sur l’Inflammation (CRI), Faculté de Médecine X. Bichat, Université Paris Cité, 75018 Paris, France

**Keywords:** inflammation, inflammatory bowel diseases, dextran sodium sulfate-induced colitis, infliximab, orexins, mucosal healing

## Abstract

Inflammatory bowel diseases are chronic inflammation of the intestinal mucosa characterized by relapsing–remitting cycle periods of variable duration. Infliximab (IFX) was the first monoclonal antibody used for the treatment of Crohn’s disease and ulcerative colitis (UC). High variability between treated patients and loss of IFX efficiency over time support the further development of drug therapy. An innovative approach has been suggested based on the presence of orexin receptor (OX1R) in the inflamed human epithelium of UC patients. In that context, the aim of this study was to compare, in a mouse model of chemically induced colitis, the efficacy of IFX compared to the hypothalamic peptide orexin-A (OxA). C57BL/6 mice received 3.5% dextran sodium sulfate (DSS) in drinking water for 5 days. Since the inflammatory flare was maximal at day 7, IFX or OxA was administered based on a curative perspective at that time for 4 days using intraperitoneal injection. Treatment with OxA promoted mucosal healing and decreased colonic myeloperoxidase activity, circulating concentrations of lipopolysaccharide-binding protein, IL-6 and tumor necrosis factor alpha (TNFα) and decreased expression of genes encoding cytokines in colonic tissues with better efficacy than IFX allowing for more rapid re-epithelization. This study demonstrates the comparable anti-inflammatory properties of OxA and IFX and shows that OxA is efficient in promoting mucosal healing, suggesting that OxA treatment is a promising new biotherapy.

## 1. Introduction

Inflammatory bowel diseases (IBDs) are chronic inflammatory disorders affecting the gastrointestinal tract with the two main forms being ulcerative colitis (UC) and Crohn’s disease. IBD etiology is not fully understood, but genetics and environment both play a significant role [1]. UC affects specifically the mucosal and submucosal layers of the colon and rectum. UC is characterized by diffuse, nonspecific inflammation that continuously damages the colonic mucosa from the rectal side, often leading to erosions and ulcers [2]. The disease activity has been associated with elevated levels of the pro-inflammatory cytokine tumor necrosis factor alpha (TNFα), which is a key player in the initiation and modulation of the inflammatory flare [3]. In recent years, several studies have found a strong association between ameliorated relapsing–remitting cycles and sustained clinical remission, together with decreased necessity for hospitalization and surgery, thus leading to the consideration of objectified mucosal healing in endoscopy (complete absence of inflammatory and ulcerative lesions), as a therapeutic goal [2]. The treatment of UC, which depends on the disease severity, involves the use of anti-inflammatory compounds, including 5-aminosalicylates, corticosteroids or immunosuppressive drugs [4]. The role of different cytokines, such as TNFα and other soluble mediators, in UC led to the development of anti-TNFα, anti-IL-12/23, anti-integrin α4β7 and Janus kinase inhibitors [5]. Infliximab (IFX) is a chimeric immunoglobulin G1 monoclonal antibody (mAb) that targets TNFα and which promotes mucosal healing and clinical remission [1]. IFX was the first mAb approved for the treatment of IBD and is widely used in the management of moderate-to-severe UC [6]. The use of IFX alone or in combination with other drugs has been a substantial advancement in UC treatment. However, many patients lose their response over time, warranting further improvements in drug therapy and the treatment strategy. The discovery of new targets for drugs represents a main challenge for UC treatment.

Orexin receptor type 1 (OX1R), a G protein-coupled receptor (GPCR) activated by orexins, might represent such a new target for innovative treatment in UC patients [7]. The role of the orexins/OXR system was initially studied in the central nervous system, in which the major physiological role appears to be as an important wakefulness-maintaining factor [8]. Additional studies performed in various peripheral organs, including the adrenal glands, kidney, cardiovascular system, reproductive tract, adipose tissue and digestive tract, indicate that orexins have numerous regulatory roles [9,10]. A recent study showed that in pathological situations, such as IBD, the ectopic expression of OX1R in the colonic epithelium is observed [11]. It is worth noting that OX1R is not expressed in the healthy colonic epithelium [12]. Moreover, a previous study demonstrated that orexin-A (OxA) has a beneficial immunomodulatory effect in a murine colitis model by binding to OX1R [11]. Lastly, colons from dextran sodium sulfate (DSS) treated mice display an inflammatory infiltrate composed of immune cells that express OX1R. Interestingly, the protective OxA effect is mediated through the inhibition of pro-inflammatory cytokines, such as TNFα and IL-6 [11]. Since several pre-clinical studies have shown the anti-inflammatory properties of OxA not only in models of IBD, but also in other chronic/acute inflammatory diseases, such as multiple sclerosis [13] and septic shock [14], it urges further studies to reinforce the view that the OxA/OX1R system may represent a promising new target in the treatment of UC.

In this context, the aim of the present study was to compare, in a model of DSS-induced colitis previously validated in our laboratory [15], the curative efficacy of OxA compared to IFX on inflammatory flare and mucosal healing. To mimic the situation of UC patients, IFX or OxA were given when colitis is maximal. Both treatments promoted mucosal healing, but OxA was found to restore more rapidly the expression of epithelial cell markers than IFX. In addition, OxA reduced efficiently plasma and gene expression of inflammation marker. These results suggest that OxA treatment could constitute a promising new biotherapy.

## 2. Results

### 2.1. Effect of DSS, IFX and OxA on Body Weight

Figure 1 shows the experimental protocol used in this study. Mice received DSS in drinking water for 5 days. On day 5, the animals were then randomly distributed into the different experimental groups. With colitis being maximal on day 7, the treatment was initiated at day 7. Mice treated with IFX received only one injection at day 7, while those receiving OxA were injected every morning for 4 days. Mice were euthanized at day 11.

As previously demonstrated, the administration of 3.5% DSS induced an important body weight loss that was maximal at day 7, thus two days after DSS removal. Body weight recovery appeared to start more rapidly for the mice receiving IFX or OxA than for untreated animals (Figure 2A). The increase in body weight gain was associated with an increase in the dietary daily intake that was similar for the DSS untreated mice and for the mice receiving IFX or OxA (Figure 2B).

At day 11, the weight loss was significantly less marked for the IFX- and OxA-treated mice (Figure 3A) compared to that for the untreated DSS mice. This effect was associated with a higher lean mass (Figure 3B), but no modification of the fat mass was measured (Figure 3C).

The administration of 3.5% DSS induced inflammation that was associated with an increased water content in the colonic fluid and thus with diarrhea. Four-day treatment with IFX or OxA decreased the water content in the colonic content when compared with that in untreated animals (Figure 4A). However, we did not observe any significant effect of the two treatments on the colon weight (Figure 4B) or length (Figure 4C), but OxA tended to increase the colon length (*p* < 0.057).

### 2.2. Effect of DSS, IFX and OxA on Inflammatory Markers

As we previously showed in a similar experimental model that mucosal healing starts at day 10, mice were euthanized at day 11. Moreover, this study showed that DSS treatment provoked a marked increase in colon MPO activity, which was associated with the mucosal and submucosal infiltration of mononuclear cells and with an increase in the colon proinflammatory cytokines (TNFα and IL-6). To evaluate the efficiency of IXF and OxA, the factors implicated in epithelial repair were evaluated. Colonic MPO enzymatic activity remained elevated for the DSS mice, but both treatments were able to bring back MPO to near its basal value at day 11 (Figure 5A). The plasmatic concentration of lipopolysaccharide-binding protein (LBP) was significantly reduced by IFX or OxA injection, but remained significantly increased compared to that in the control mice (Figure 5B). Both treatments were able to reduce pro-inflammatory plasma concentrations of TNFα and IL-6 (Figure 5C,D), but OxA treatment was significantly more efficient in restoring the IL-6 plasma basal concentration and reduced, more importantly, the TNFα plasma concentration when compared to those with IFX treatment (Figure 4D).

### 2.3. Histological Changes Induced by DSS and Effects of IFX and OxA Treatments

DSS treatment induced severe histological damage. At day 11, as expected, DSS-treated mice, when compared to control mice, showed massive inflammation, as evidenced by the mucosal and submucosal infiltration of mononuclear cells and infiltration of neutrophils located at the edges of ulcerated areas. The presence of polynuclear cells was evidenced by the nucleus shape of cells that were found in ulcerated areas (indicated by black arrows Figure 6A). Treatment with IFX or OxA induced a major reduction in this infiltration. DSS-induced inflammation caused crypt disappearance associated with distension of the remaining crypts. An evaluation of the colonic length of well-oriented epithelial crypts showed that DSS induced an important increase in the colonic length (149 ± 3 µm for the control vs. 391 ± 11 µm for the DSS group), indicative of epithelial hyperproliferation for colon epithelial repair. Colon crypt length was more significantly reduced by OxA than IFX (331 ± 9 µm vs. 262 ± 11 µm for IFX and OxA respectively). Moreover, the mucin-secreting cell pool that was depleted by DSS was largely restored by IFX and OxA treatments (Figure 6B). IFX and OxA treatments were able to allow for an accelerated re-epithelization, and epithelial repair appeared more efficient when the mice received OxA as compared to IFX.

### 2.4. Effect of DSS, IFX and OxA on Colonic mRNA Expression of Different Genes Involved in Colonic Inflammation and Healing

As previous described, our data show that DSS-induced inflammation increased the expression of the inflammatory cytokines *Tnf-α* and *Il-6* and the regulatory cytokine *Il-10*, whereas the expression of *Il-13* was decreased (Table 1). IFX or OxA treatments markedly diminished the expression of genes encoding *Tnf-α*, *Il-6* and *Il-10* when compared with that in DSS-treated animals not receiving IFX of OxA, thus suggesting a reduction in the ongoing inflammation intensity (Table 1). DSS-induced inflammation also modulated the expression of genes involved in epithelial repair, namely *Il-15*, which was decreased, and *Igf-1* and *Il-22*, for which expression was increased. IFX and OxA treatments restored the basal expression of *Il-15*, while *Igf-1* was only partly restored, reflecting ongoing colonic repair. DSS treatment decreased the expression of the tight-junction marker occludin (*Ocln*) and of the mucin-secreting cell markers *Muc2* and *Klf4*. Inversely, at day 11, DSS did not reduce the expression of the tight-junction marker *Tjp1*. IFX and OxA treatments allowed for an almost complete restoration of *Ocln*, *Muc2* and *Klf4* gene expression. However, as previously shown, OxA treatment after colitis induction increase the expression of tight-junction *Tjp1* expression when compared to that with colitis induction without treatment. Improvement in the barrier function after treatment was associated with increased expression of *Ocln* and goblet cell markers, as well as the restoration of epithelial-repair-modulating factors.

## 3. Discussion

The results of the present study show a reduction in inflammation, as evidenced by the reduction of TNFα and IL-6, measured as gene expression and plasma levels. In addition, an accelerated re-epithelization and epithelial repair are supported by the reduction in the colon crypt length, such reduction being more important when mice received OxA in comparison with IFX. These results are compatible with the view that OxA given from a curative perspective is efficient in experimental colitis for the reduction of several parameters known to be increased in inflammatory bowel diseases. Furthermore, OxA was able to restore the expression of epithelial cell markers and mucosal healing after colitis. These effects were associated with the OxA capacity to slow down lean mass loss that parallel colitis induction and to reduce the water content in the colonic fluid. Of note, the beneficial effects of OxA on the different parameters of inflammation and healing were apparently more marked than the ones observed using the anti-TNFα IFX.

The experimental colitis induced by DSS displays analogies with the anomalies observed in UC [16,17]. UC is a life-long disease that affects the mucosal and submucosal layers of the colon and rectum. UC has been associated with elevated levels of pro-inflammatory cytokines, including TNFα, which is a key player in the initiation and modulation of chronic inflammation. Usually, the first treatment used to treat UC is topical 5-aminosalicylic acid (5-ASA) drugs. UC patient with more extensive or severe disease can be treated with a combination of oral and topical 5-ASA drugs, with or without corticosteroids. When a severe UC patient needs to be hospitalized for treatment, they usually receive intravenous steroids and if refractory, calcineurin inhibitors (cyclosporine, tacrolimus) or IFX [3]. Patients are then maintained on the appropriate medications to allow for remission. However, 30% of the UC patients in their lifetime will need a surgical intervention [18]. IFX is a chimeric (human/mice) monoclonal antibody that has great affinity for the soluble and membrane forms of TNFα inducing an anti-inflammatory impact by reducing the TNFα level [19]. IFX was the first successful treatment for IBD patients [20], but many other TNFα monoclonal antibodies have been developed thereafter. These pharmacological approaches are believed to play a protective role and promote intestinal healing. Indeed, IFX therapy has been shown to reduce the number of patients needing colectomy [21]. However, about one-third of patients do not respond to anti-TNFα drugs [22], and many patients are at risk over time to lose response or develop intolerance and adverse events [23]. These studies support the view that improvements in drug therapy and new innovative strategies are needed. In that context, the identification of new targets for the treatment of UC, as well as compounds active on these targets, represents a major challenge.

Orexins (OxA and OxB) are hypothalamic neuropeptides that are synthetized in the central nervous system and are involved in many physiological processes. The most important targets of orexins are present in the central nervous system, but they also play a role in various peripheral organs, including the intestine. Orexins trigger their central and peripheral biological effects by interacting with two members of the GPCRs, namely orexin receptor-1 (OXIR) and orexin receptor-2 (OX2R). GPCRs belong to a large family of seven-transmembrane cell surface receptors that are likely to represent such innovative targets. Several studies indicated that GPCRs through interactions with cannabinoid receptors [24], neuropeptide receptors [25], histamine receptors [26] and chemokine receptors [27] may likely play therapeutic roles in IBD.

Regarding orexin receptors, a previous study [11] showed that OX1R is expressed in the inflamed human epithelium of UC patients and in the mucosa of Crohn’s disease patients, but not in the normal colon mucosa. Moreover, OX1R is also expressed in the DSS-induced colitis mouse model, whereas this receptor, as in the human normal mucosa, is not expressed in the mouse normal mucosa [11].

The studies of the orexin signaling pathways have shown that the binding of OxA to OX1R activates the heterotrimeric Gq protein via GTP/GDP exchange, which leads to the dissociation and activation of the αq subunit, thus inducing the activation of phospholipase C (PLC) (Figure 6). PLC catalyzes the production of inositol-1,4,5-trisphosphate (IP3) and diacyl glycerol, then leading to transient Ca^2+^ release in the cytoplasm (Figure 7), with such release being responsible for the activation of several signaling pathways, such as MAPK (mitogen-activating protein kinases)-Erk1/2 (extracellular signal-regulated kinases), cAMP, JNK (c-JUN N-terminal kinases) and PI3K (phosphoinositide 3-kinase)-Akt. Such cascades of activation following treatment with orexins were observed. In the context of acute/chronic inflammation, the increase in intracellular calcium induced by OxA inhibits the activation of NFκB, thus blocking its translocation to the nucleus and thus blocking the expression of many pro-inflammatory cytokines encompassing TNFα (Figure 7). It should be noted that immune cells in UC express OX1R, and OxA inhibits the secretion of pro-inflammatory cytokines by these cells [11].

An evaluation of the mechanism of action of IFX in IDB showed that the mechanisms implicated in UC remission are different from those reported for OxA. TNFα has two receptors, TNFR1 and TNFF11, which have distinct intracellular signaling pathways [28]. Several studies reported that the efficacity of IFX in humans is not related exclusively to TNFα [29]. Indeed, it appears that the major effect of IFX is related to the interruption of the anti-apoptotic signaling pathway through an interaction with expression of the membrane-bound form of TNFα (mTNFα) on monocytes [30]. This latter interaction involves TNFRII that is expressed on lamina propria T cells and which induces T cell apoptosis [31,32]. A second mechanism of IFX involves the Fc region of the compound that induces M2-type wound-healing macrophages that can complete mucosal healing (Figure 7) [33]. The TNFR1 mainly binds soluble TNFα, and TNFR11 binds mTNFα [28]. TNFRI, which is associated with TRADD, promotes inflammation and is expressed ubiquitously. Recent studies suggest that TNFα neutralization is not the major IFX mechanism of action, thus supporting the fact that the activation of the NF-κB transcription factor family is not involved in the resolution of inflammation and mucosal healing mediated by IFX [34]. Moreover, it has been shown in a mouse model that the anti-inflammatory effects of IFX are not related to the prevention of TNFα production and that IFX is not able to block NF-κB signaling [35].

Although the treatment with IFX and OxA results in comparable effects on numerous parameter characteristics of mucosal inflammation and subsequent healing, the present study demonstrates that OxA was even more efficient than IFX in reducing inflammation markers (TNFα and IL-6 plasma level) and inducing colon mucosa healing, as indicated by the significant more important reduction in the epithelial crypt, suggesting that OxA and IFX may have different mechanisms of action. To increase the efficiency of such treatments, the development of new approaches, such as nano approaches, would be a good alternative for the delivery of treatment resolving inflammation, especially when they need to be injected [36].

In conclusion, our study shows that OxA, as IFX, was able, from a curative perspective, to restore mucosa integrity and the intestinal barrier. The OXR1/OxA system is known to be involved in inhibition of the secretion of various pro-inflammatory cytokines, such as TNFα and IL-6. The inhibitory impact of OxA is known to be mediated through the inhibition of NF-κB activation resulting of intracellular Ca^2+^ release, via the phospholipase C, induced by the activation of OX1R and its associated Gq protein [37], but additional works, outside the objective of the present work, are obviously needed to confirm this hypothesis. The inhibition of the NF-κB activation would then allow for a reduction in inflammatory cytokine secretion, then possibly inducing the observed anti-inflammatory effect [38]. It would be interesting to evaluate the effect of OxA on the mucosa-adherent microbiota, as recent studies support the fact that bacterial species associated with remission after anti-cytokine therapy do have anti-inflammatory effects [39,40]. Finally, it is worth noting that, according to the elements of the discussion above, the mechanisms of action of OxA allowing for a reduction in the inflammatory flare would be different than the ones involved in the IFX effects. Even if this study is purely experimental, OxA represents a new innovative molecule having putative therapeutical interest in the treatment of UC, prompting additional experimental studies and then, hopefully, clinical studies.

## 4. Materials and Methods

### 4.1. Animals

Male C57BL/6 OlaHsd mice (7 weeks) were acclimated for 1 week with free access to standard mouse chow and tap water. They were housed in collective cages for the duration of the study in an air-conditioned room with a controlled temperature (22 ± 1 °C), humidity (65–70%) and reversed day/night cycle (12 h dark from 9 h to 21 h light 21 h to 9 h). The study was performed according to the European directive for the use and care of laboratory animals and received the approval of the French Government (registration number: APAFIS#32545-202107151059340).

### 4.2. Experimental Design

The mice were then fed a standard AIN-93M diet with an energy distribution of 14% of total energy as milk protein, 74% carbohydrate (80% starch, 20% sucrose) and 10% soybean oil (Table 2). The control of colonic inflammation kinetics was controlled in this model as described [15]. Healthy controls (untreated mice at day 0, *n* = 12) received fresh tap water. DSS-treated mice (*n* = 40) received 3.5% (*w*/*v*) DSS (36,000–50,000 MW, MP Biomedicals, Illkirch-Graffenstaden, France) in drinking water for 5 d, thus from day 1 to day 5, to induce an acute episode of colitis (Figure 1). Drink and food were provided *ad libitum*. On day 5, the animals were then randomly distributed into the different experimental groups. The reverse cycle allowed us to perform the injections in the morning before the start of the nocturnal period that induces food intake.

With the colitis being maximal on day 7, IFX was given as one single injection at day 7 via intraperitoneal (i.p.) injection at a concentration of 7 µg/g of body weight, and OxA was given for 4 days (GL Biochem, Shanghai, China) by diluting this peptide in phosphate buffer saline (PBS). Further, 1 μmol of OxA per kg of body weight was i.p. administrated daily (at 09:00) in accordance with a previous experiment [41], showing that OxA did not have any deleterious effect on the colon mucosa of healthy control mice. The animals had free access to food and drinking water throughout the experiment. The animals were euthanized in all cases 4 days after the start of the treatment.

### 4.3. Body Composition

The body composition (fat mass and lean mass) was measured at the beginning and at the end of the study via dual energy X-ray absorptiometry (DEXA), using a Lunar PIXImus densitometer (DEXA-GE PIXImus). The stability of the device was controlled by the measurement of a phantom before each session. The mice were anesthetized via isoflurane inhalation during the measurement. Analysis of the images was performed with the software provided with the device (Lunar PIXImus v2.10), using auto-thresholding.

### 4.4. Tissue Collection

Animals were anesthetized via the inhalation of isoflurane. Blood was withdrawn via intracardiac puncture, and tissues were collected. Blood was collected in EDTA tubes, and plasma was frozen and stored at −80 °C. The entire colon was removed, measured and weighed. Whole luminal colonic content was then removed for water content measurements via desiccation. The colon was divided into 4 segments. One segment was harvested for RNA analysis and immediately frozen in TRIzol^®^ Reagent (ThermoFisher Scientific, Waltham, MA, USA) and stored at −80 °C until further analysis. Two other segments were immediately frozen in liquid nitrogen and stored at −80 °C for myeloperoxidase (MPO) and protein expression assays. One segment of the distal colon was fixed in 4% buffered formaldehyde for histological analysis.

### 4.5. Quantification of Gene Expression via Real-Time Polymerase Chain Reaction (qRTPCR)

The samples were kept at −80 °C, and they were unfrozen just before total RNA extraction. Total RNAs were extracted using TRIzol reagent, after homogenization using a Tissue lyser (Qiagen SAS, Courtaboeuf, France), and RNA concentrations in samples were measured with a NanoDrop ND-1000 UV-Vis spectrophotometer. RNA was purified using an RNeasy Minikit (Qiagen) and DNase I treatment. Total RNA (0.4 μg) in a final volume of 10 μL was reverse transcribed using a high-capacity cDNA archive kit protocol (Life Technology, Courtaboeuf, France). qRT-PCR was performed with Fast SYBR Green MasterMix (Applied Biosystems, ThermoFisher Scientific, Illkirch, France), using gene-specific primers (sequences available on demand) and the StepOne Real-Time PCR system (Applied Biosystems, ThermoFisher Scientific, Illkirch, France) as previously described [15]. Gene expression was determined using the 2^−ΔΔCT^ formula, where ΔΔCT = (CT target gene—CT reference gene) using *Hprt* as the house-keeping gene.

### 4.6. Histological Analysis

After fixation, colonic sections (4 µm) were stained with hematoxylin and eosin using the Histo Pathology High Precision (H2P2) platform. Histological colon damage and repair were evaluated through microscopic examination. Images were digitalized using a slide scanner, and colonic length of well-oriented epithelial crypts was determined through image analysis using NDP.view2Plus. Periodic Acid Schiff staining was used to visualize mucus-producing cells on 4 μm transversal colon sections counterstained with hematoxylin.

### 4.7. Determination of Local and Systemic Inflammatory Markers

Intestinal tissue was assayed for MPO activity, used as a neutrophil infiltration marker. Activity analysis was performed using an O-dianisidine dihydrochloride assay as previously described [42]. Plasma TNFα and IL-6 concentrations were determined using a sandwiched ELISA kit (Thermo Scientific, Courtaboeuf, France) according to the manufacturer’s instructions. Plasmatic concentrations of lipopolysaccharide-binding protein (LBP) were determined with a commercial solid-phase sandwich enzyme-linked immunosorbent assay from Abcam (Mouse LBP SimpleStep ELISA kit), Paris, France.

### 4.8. Statistical Analysis

Results were expressed as means ± SEMs and compared using a one-way analysis of variance (ANOVA) and a Tukey multiple comparison test or *t*-test, which was also used to assess the effect of treatments or differences between two treatments. Significance was established at *p* < 0.05. All statistical analyses were performed using Prism^®^ Version 6.05 (GraphPad Software Inc., San Diego, CA, USA).

## Figures and Tables

**Figure 1 ijms-24-09554-f001:**
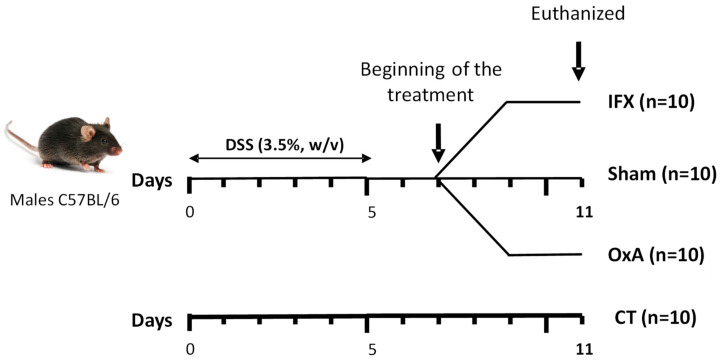
Schematic representation of the experimental design. The control (CT) mice did not receive any treatment. Mice were treated with dextran sodium sulfate (DSS) for 5 days. DSS-treated mice received either nothing (Sham), infliximab (IFX) only at day 7 via intraperitoneal (i.p.) injection at a concentration of 7 µg/g of body weight or orexin-A (OxA) for 4 days via i.p. injection at a concentration of 1 μmol/kg of body weight. Mice were euthanized at day 11.

**Figure 2 ijms-24-09554-f002:**
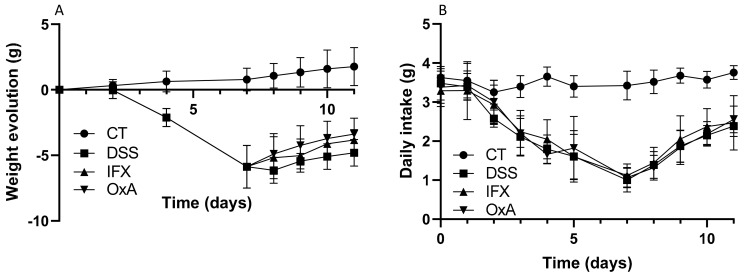
Weight gain evolution (**A**) and daily intake (**B**) as a function of time values are means ± SEM (*n* = 10). CT: control; DSS: dextran sodium sulfate; IFX: infliximab; OxA: orexin-A.

**Figure 3 ijms-24-09554-f003:**
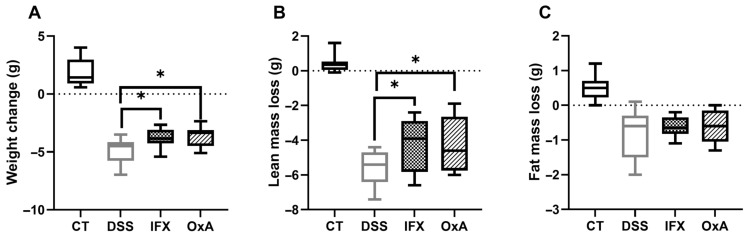
Weight evolution after 11 days (**A**), lean mass loss (**B**) and fat mass loss (**C**); values are means ± SEMs (*n* = 10). A *t*-test was used to compared DSS-untreated animals to IFX- or OxA-treated animals (* *p* < 0.05). Lean and fat mass were evaluated using a Lunar PIXImus densitometer. CT: control; DSS: dextran sodium sulfate; IFX: infliximab; OxA: orexin-A.

**Figure 4 ijms-24-09554-f004:**
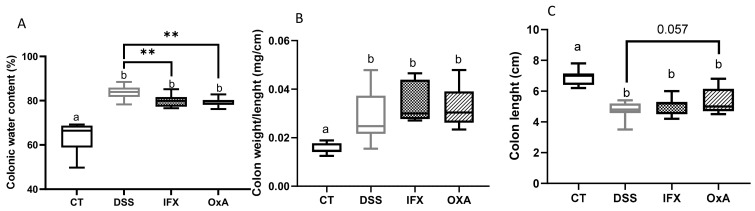
Effect of DSS, IFX and OxA at day 11 on feces water content (**A**), colon weight/length (**B**), and (**C**) colon length. Values are means ± SEM (*n* = 10). Values in the different groups were compared using a one-way analysis of variance (ANOVA) and a Tukey multiple comparison test. Values with different letters are significantly different (*p* < 0.05). A *t*-test was used to compared DSS-untreated animals with IFX- or OxA-treated animals (** *p* < 0.01). CT: control; DSS: dextran sodium sulfate; IFX: infliximab; OxA: orexin-A.

**Figure 5 ijms-24-09554-f005:**
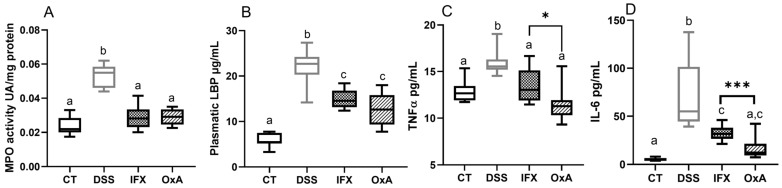
Effect of DSS, IFX and OxA at day 11 on the plasma inflammation markers MPO activity (**A**), LBP (**B**), TNFα (**C**), and IL-6 (**D**) plasma levels. Values are means ± SEMs (*n* = 10). Values were compared using a one-way analysis of variance (ANOVA) and a Tukey multiple comparison test. Values with different letters are significantly different (*p* < 0.05). A *t*-test was also used to assess the difference between the two treatments (* *p* < 0.05, *** *p* < 0.001). CT: control; DSS: dextran sodium sulfate; IFX: infliximab; OxA: orexin-A; MPO: myeloperoxidase; LBP: lipopolysaccharide-binding protein; TNFα: tumor necrosis factor alpha; IL-6: interleukin-6.

**Figure 6 ijms-24-09554-f006:**
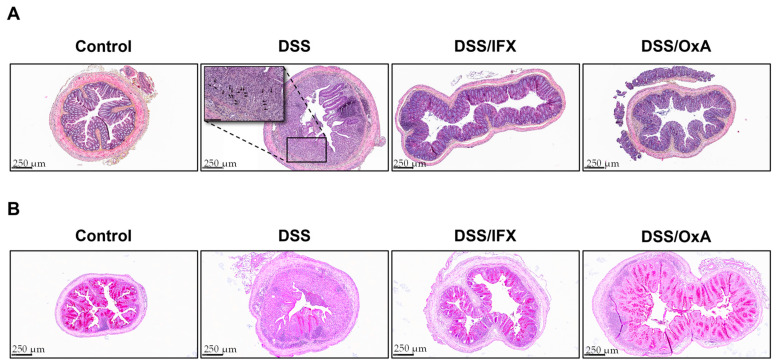
Colons obtained at day 11 were fixated, and colonic sections (4 µm) were stained with hematoxylin and eosin (**A**). Periodic Acid Schiff staining was used to visualize mucus-producing cells based on transversal colon sections counterstained with hematoxylin (**B**). CT: control; DSS: dextran sodium sulfate; IFX: infliximab; OxA: orexin-A.

**Figure 7 ijms-24-09554-f007:**
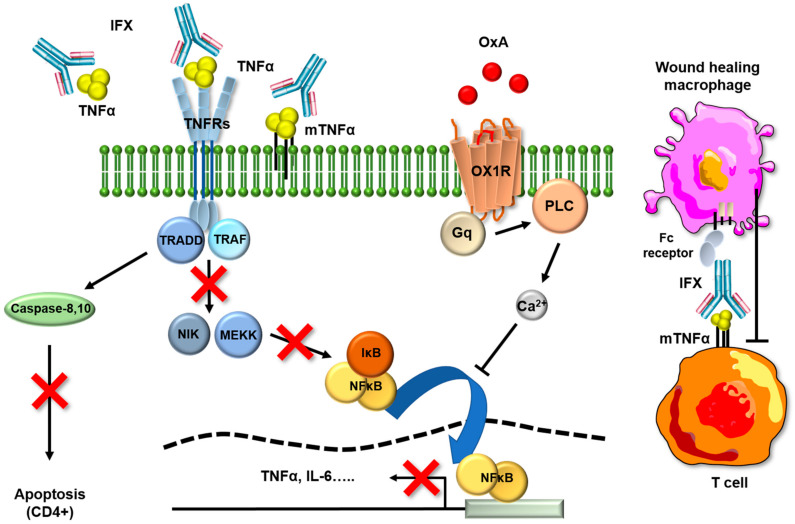
Oxa and IFX main signaling pathways involved in anti-inflammatory effects. The orexin/receptor system may trigger the anti-inflammatory functions though the inhibition of NF-κB. An anti-apoptotic signal is regulated by mTNFα, and this mechanism is inhibited by anti-TNFs inducing, consequently, the apoptosis of CD4^+^ cells. Anti-TNFs also bind to mTNFα on activated T cells, and consequently, the Fc part of the antibody is recognized by the Fc receptors, thus triggering the wound-healing macrophages. TNFα: tumor necrosis factor alpha; IL-6: interleukin-6; IFX: infliximab; TNFRs: tumor necrosis factor alpha receptors; mTNFα: membrane-bound from of tumor necrosis factor alpha; TRADD: tumor necrosis factor alpha receptor type 1 associated death domain protein; TRAF: tumor necrosis factor alpha receptor-interacting protein 1; NF-κB: nuclear factor Kappa B; NIK: nuclear factor Kappa B-inducing protein; MEKK: mitogen-activated protein/ERK kinase; IKB: inhibitor of nuclear factor kappa B; OX1R; orexin receptor 1; Gq: G protein; PLC: phospholipase C.

**Table 1 ijms-24-09554-t001:** Effect of DSS, IFX and OxA on colonic gene expression.

Parameters	CT	DSS	IFX	OxA
Inflammatory markers				
*Tnf-a*	1.09 ± 0.16 ^a^	3.55 ± 0.30 ^b^	1.86 ± 0.28 ^a^	1.93 ± 0.36 ^a^
*Il-6*	1.03 ± 0.12 ^a^	11.17 ± 0.70 ^b^	5.21 ± 1.11 ^c^	7.53 ± 0.65 ^c^
*Il-10*	1.07 ± 0.07 ^a^	11.44 ± 1.26 ^b^	4.81 ± 0.63 ^c^	3.59 ± 0.57 ^a,c^
*Il-13*	1.08 ± 0.08 ^a^	0.46 ± 0.05 ^b^	1.17 ± 0.08 ^a^	0.91 ± 0.13 ^a^
Tight-junction protein				
*Tjp1*	1.01 ± 0.16 ^a,b^	0.60 ± 0.05 ^a^	1.04 ± 0.07 ^a,b^	1.15 ± 0.22 ^b^
*Ocln*	1.03 ± 0.07 ^a^	0.48 ± 0.04 ^b^	1.02 ± 0.17 ^a^	1.06 ± 0.23 ^a^
*Cldn2*	1.06 ± 0.15	1.51 ± 0.30	1.83 ± 0.16	1.35 ± 0.32
Goblet cell markers and mucins				
*Muc2*	0.99 ± 0.09 ^a^	0.37 ± 0.04 ^b^	1.02 ± 0.15 ^a^	0.87 ± 0.12 ^a^
*Klf4*	1.00 ± 0.06 ^a^	0.42 ± 0.06 ^b^	0.74 ± 0.09 ^a^	0.74 ± 0.08 ^a^
Epithelial-repair-modulating factors				
*Igf1*	1.09 ± 0.18 ^a^	4.59 ± 1.05 ^b^	2.10 ± 0.34 ^a^	2.15 ± 0.25 ^a^
*Il-15*	1.02 ± 0.10 ^a^	0.39 ± 0.05 ^b^	1.05 ± 0.21 ^a^	1.24 ± 0.22 ^a^
*Il-22*	1.02 ± 0.24 ^a^	3.05 ± 0.36 ^b^	2.81 ± 0.43 ^b^	2.66 ± 0.59 ^a,b^

mRNA expression (expression in arbitrary units) of the different genes (expression relative to *Hprt*) in colon segments that were collected at day 11. Values are means ± SEMs (*n* = 10). Means that are significantly different (*p* < 0.05) according to the Tukey multiple comparison test have different letters. *Tnf-α*: tumor necrosis factor alpha; *Il*: interleukin; *Tjp1:* tight junction protein; *Ocln:* ocludin; *Cldn2:* claudin 2; *Muc2:* mucin 2; *Klf4:* kruppel-like factor 4; *Igf1:* insulin growth factor 1.

**Table 2 ijms-24-09554-t002:** Composition of the experimental diet.

Ingredients (g/kg of Diet)	P14
Acid casein (Armor Protéines^®^, ref. 139860)	112
Whey protein (Armor Protéines^®^, Protarmor 80, ref. 139805)	28
Corn starch	622.4
Sucrose	100.3
Soybean oil	40
Alpha cellulose	50
AIN 93M mineral mix	35
AIN 93M Vitamins	10
Choline	2.3
Metabolizable energy, kJ/g	14.5

## Data Availability

No new data were created or analyzed in this study. Data sharing is not applicable to this article.

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
