# Peer review of "Efficiency of Orexin-A for Inflammatory Flare and Mucosal Healing in Experimental Colitis: Comparison with the Anti-TNF Alpha Infliximab"

_ijms, 2023, doi:10.3390/ijms24119554_

Round 1

Reviewer 1 Report

The study “Efficiency of Orexin-A on inflammatory flare and mucosal healing in experimental colitis: comparison with anti-TNF alpha infliximab” by Anne Blais et al. is a well-designed study with the goal of identifying a possible new treatment option for colitis. Results from experiments suggest that the single dose of orexin tested is as effective as the single dose of IFX tested in reducing the colonic symptoms caused by experimentally-induced colitis.

Major concern: The effects of OxA on treating a similar experimental model of colitis (DSS) has already been published by these authors (Messal et al. 2018). While this paper is referenced in the methods section and in the introduction, the major results from that paper (that in fact OxA is effective at treating inflammatory symptoms) is left out of the current manuscript. As such, the results of the current study seem more novel than they are. Significantly more references should be made to the results of the previously published study in terms of treatment effectiveness. Results of the current study (comparing treatment effectiveness between OxA and IFX) diminutive in light of this previous publication.

Introduction

Line 51 – minor edit: consider replacing “of” with “on”

Line 52 – minor edit: consider changing “involved” to “involves”

Line 66 – it is unclear what is meant by the phrase “major biological appears”. Please edit or explain.

Line 67 – minor edit: consider removing the word “Moreover” at the beginning of the sentence since it is used 2 sentences later.

Line 69-70 – where is pathological OX1R expression observed in IBC?

Line 71 – minor edit: replace “to note” with “noting”

Line 71 – reverse the order of the words “colonic healthy”

Line 72 – minor edit: replace “demonstrates” with “demonstrated”

Line 73 – define DSS since the abbreviation has not been used yet

Line 79 – minor edit: insert the word “the” between “ that OxA/OX1R”

Line 81 – remove “dextran sodium sulfate”

Line 83 – minor edit: replace the word “shown” with “shows”

Results

Line 88 – remove the bold font and period in “Figure 1.”

Methods

It is mentioned that a single dose of OxA was used (1umol/kg body weight) in accordance with previous experiments. However, the “previous experiments” referenced utilize a different dose (0.22umol/kg body weight) and with a different protocol (OxA was administered prior to DSS). An explanation for the discrepancy should be provided.

Results

Line 98  - the statement “appeared to start more rapidly for the mice receiving IFX or OxA”. This is not supported by statistics. A 2-way RM ANOVA should be run on the data in Figure 2, with time and treatment as factors.

Discussion

Line 192-195 – specific results should be referenced here.

Major concern: There are many typographical and English grammar errors making it almost unreadable. A thorough edit should be done. I made edits in the introduction, but there were too many to continue this practice throughout the manuscript.

Reviewer 2 Report

1.      In figure 5, authors show increase in MPO activity. Is it the increased enzyme activity or increase enzyme expression due to higher recruitment of cells that is responsible for this inflammatory phenotype?

2.      Figure 6- What type of immune cells are infiltrating this tissue during the time course should be investigated. The difference between early and late immune landscape after DSS treatment, will add more value to this manuscript.

3.      Whether blocking Tnf-α, Il-6, and Il-10 could rescue this phenotype should be investigated.  

4.      Add type of test performed to analyze significance in figure1,2,3

5.      In figure 5 the author should perform Anova instead of T-test.

6.      Add details of sample collection time point in figure legend in figure 4,5,6

Moderate editing of English language required.

Reviewer 3 Report

In this study, Anne Blais et al. studied the anti-inflammatory properties between orexin-A (OxA) and the anti-TNFα Infliximab in a mouse model of chemically induced colitis. The authors conclude that OxA as IFX was able to restore mucosa integrity and intestinal barrier, and OxA represents a new innovative molecule having a putative therapeutical interest in the treatment of ulcerative colitis (UC). The study question is valid and explores new target in the treatment of UC. However, anti-inflammatory effect of Orexin-A in ulcerative colitis has been previously described in literature. Authors state that this is a comparative study between Orexin-A and IFX but it is not clear if Orexin-A works better than IFX or if both have similar efficacy after reviewing study findings. Authors may want to address this more in the results and discussion section. Authors may also want to add study limitations.

Reviewer 4 Report

This original research article evaluates the efficiency of orexin-A on inflammatory flare and mucosal healing in experimental colitis compared with infliximab. The topic is relevant and up-to-date, but certain deficiencies identified in both content and form need to be addressed based on the specific recommendations below:

Shape suggestions

Please review the template and instructions for authors and modify the first page on organizing author data accordingly.

Please revise the font size and text font in abstract because they are not uniform.

Abbreviations are treated separately in the abstract and in the main text. If it is mentioned only once, no abbreviation is necessary (L19- CD). Please revise the whole manuscript.

Any abbreviations used in tables or figures must be explained in the form of a legend below these structures. Please review the whole manuscript from this point of view (especially table 1).

L248; L266- No blank space is required between paragraphs from the same section/subsection. Please revise this aspect throughout the manuscript.

The conclusion section comes at the end of the text as the last chapter. The materials and methods section should be numbered as appropriate (also the subsections).

 Content suggestions

The conclusion part of the abstract should be improved in terms of outcomes and what future research directions this research may refer to.

It is advisable to detail the relevance of gut microbiota manipulation (as an important element in the pathophysiologic mechanism of IBD) in inflammatory bowel disease. I suggest checking and referring to: PMID: 34203609.

The aim of the paper should be improved from the perspective of describing the contribution to the field under analysis and the elements of scientific novelty presented, especially since it is not the first manuscript evaluating these aspects.

L145 - the results section should not contain references to bibliographic resources. References and comparisons with other studies/information/techniques should be made in the discussion section.

It is important to mention the impact of nanoapplications on solving unmet needs in conventional therapies for different complex pathologies. It is important to understand why nanoapproaches are needed and what issues they could address because they also present disadvantages. I suggest checking and referring to the next model of nanomanagement: https://authors.elsevier.com/c/1gvan5aZoHgBmZ / (PMID: 37031724).

In the last paragraph of the Discussion section, it is advisable to detail the strengths but especially the limitations of your study and to what extent they could be resolved in view of future research directions.

Round 2

Reviewer 1 Report

The concerns regarding the methodology were adequately addressed. 

There are still many mistakes in regards to English grammar. It is still difficult to read. 

Author Response

Reviewer 1
We warmly thank the reviewer for final comment

There are still many mistakes in regards to English grammar. It is still difficult to read.

The manuscript has been again carefully edited and read by a native English.

Reviewer 2 Report

In figure 5, authors show increase in MPO activity. Is it the increased enzyme activity or increase enzyme expression due to higher recruitment of cells that is responsible for this inflammatory phenotype?

Response: It is an increase of the enzyme activity. The information has been added in the revised version of the manuscript (Line 169).

I do not see any such information in Line 169. I found some information in line 145. However without any experimental proof the answer doesn’t address whether increased MPO activity is sufficient to drive this phenotype or this is due to higher recruitment of neutrophils itself as claimed in fig 6A, which will bring more MPO to the site. Author should acknowledge both possibilities in the discussion if this investigation is out of scope of the current submission. 

Figure 6- What type of immune cells are infiltrating this tissue during the time course should be investigated. The difference between early and late immune landscape after DSS treatment, will add more value to this manuscript.

Response: We did not evaluate in details the infiltrating immune cells but we indicated that neutrophils are located at the edges of ulcerated areas. The information was added. Early and late immune landscape after DSS treatment have already been studied by our team (Vidal-Lletjós S et al 2019 World J Gastroenterol and Nutrients). This is why we choose to study the effect of IFX and OxA at 11 days. At this time, point inflammation remains present but epithelial repair occurs. Modifications were done in the introduction (line 83-89) and at the beginning of the results section line 92-96 and 136-137.

On what basis author is claiming the presence of neutrophils in these tissues? Author should submit representative images or perform immunostaining of these samples in order to claim such information.

Author Response

Reviewer 2

We warmly thank the reviewer for taking the time to review our manuscript and for offering several insightful suggestions.

All modifications were written in red in the modified manuscript. All page and line numbers corresponded to the revised manuscript.

In figure 5, authors show increase in MPO activity. Is it the increased enzyme activity or increase enzyme expression due to higher recruitment of cells that is responsible for this inflammatory phenotype?

Response: It is an increase of the enzyme activity. The information has been added in the revised version of the manuscript (Line 169).

I do not see any such information in Line 169. I found some information in line 145. However without any experimental proof the answer doesn’t address whether increased MPO activity is sufficient to drive this phenotype or this is due to higher recruitment of neutrophils itself as claimed in fig 6A, which will bring more MPO to the site. Author should acknowledge both possibilities in the discussion if this investigation is out of scope of the current submission. 

Response: MPO activity is used in this study as by others (Peter P Bradley et al, 1982, Journal of Investigative Dermatology), (Mullane, K.M. et al, 1985, J. Pharmacol. Methods) (Lan A et al, 2015, J Nutri Biochem), (Lan A et al, 2016, Am J Physiol Gastrointest Liver Physiol). MPO is an enzyme found in granulated cells and was previously shown to be a reliable index of neutrophil accumulation in tissues. A sentence was added line 144 to 148.

Figure 6- What type of immune cells are infiltrating this tissue during the time course should be investigated. The difference between early and late immune landscape after DSS treatment, will add more value to this manuscript.

Response: We did not evaluate in details the infiltrating immune cells but we indicated that neutrophils are located at the edges of ulcerated areas. The information was added. Early and late immune landscape after DSS treatment have already been studied by our team (Vidal-Lletjós S et al, 2019, World J Gastroenterol and Nutrients). This is why we choose to study the effect of IFX and OxA at 11 days. At this time, point inflammation remains present but epithelial repair occurs. Modifications were done in the introduction (line 83-89) and at the beginning of the results section line 92-96 and 136-137.

On what basis author is claiming the presence of neutrophils in these tissues? Author should submit representative images or perform immunostaining of these samples in order to claim such information.

Response: The presence of polynuclear cells (i.e polymorphonuclear cells or granulocytes) is evidenced by the nucleus shape of cells that are found in ulcerated areas (indicated by black arrows). Given the context, it is unlikely that these cells are eosinophils or basophils. These observations corroborate the measurement of MPO activity which increase is a marker of neutrophil infiltration. Figure 6 have been modified and a sentence was added line 172-173.

Reviewer 4 Report

The authors have significantly improved the paper based on the suggestions received.

Author Response

Reviewer 4

We warmly thank the reviewer for final comment.